# Brain-inspired predictive coding dynamics improve the robustness of deep neural networks

**Bhavin Choksi**[*]
CerCo CNRS, UMR 5549 &
Université de Toulouse
bhavin.choksi@cnrs.fr

**Milad Mozafari**[*]
CerCo CNRS, UMR 5549 &
IRIT CNRS, UMR 5505
milad.mozafari@cnrs.fr

**Callum Biggs O'May**
CerCo CNRS
UMR 5549

**Benjamin Ador**
CerCo CNRS
UMR 5549

**Andrea Alamia**
CerCo CNRS
UMR 5549

**Rufin VanRullen**
CerCo CNRS, UMR 5549 &
ANITI, Université de Toulouse
rufin.vanrullen@cnrs.fr

## Abstract

Deep neural networks excel at image classification, but their performance is far less robust to input perturbations than human perception. In this work we address this shortcoming by incorporating brain-inspired recurrent dynamics in deep convolutional networks. We augment a pretrained feedforward classification model (VGG16 trained on ImageNet) with a "predictive coding" strategy: a framework popular in neuroscience for characterizing cortical function. At each layer of the hierarchical model, generative feedback "predicts" (i.e., reconstructs) the pattern of activity in the previous layer. The reconstruction errors are used to iteratively update the network's representations across timesteps, and to optimize the network's feedback weights over the natural image dataset–a form of unsupervised training. We demonstrate that this results in a network with improved robustness compared to the corresponding feedforward baseline, not only against various types of noise but also against a suite of adversarial attacks. We propose that most feedforward models could be equipped with these brain-inspired feedback dynamics, thus improving their robustness to input perturbations.

## 1 Introduction

Recent studies have stressed the importance of feedback connections in the brain [1, 2], and have shown how artificial neural networks can take advantage of such feedback for various tasks such as object recognition with occlusion [3], or panoptic segmentation [4]. Feedback connections convey contextual information about the state of the higher layers down to the lower layers of the hierarchy; this way, they can constrain lower layers to represent inputs in meaningful ways. In theory, this could make neural representations more robust to image degradation [5]. Merely including feedback in the pattern of connections, however, may not always be sufficient; rather, it should be combined with proper mechanistic principles. To that end, we employ a predictive coding framework (supported by ample neuroscience evidence [6, 7, 8, 9, 10]) to build a large-scale hierarchical network with both feedforward and feedback connections that can be trained using backpropagation. Several prior studies have explored this interesting avenue of research [11, 12, 13, 14], but with important differences with our approach (see Section 2). We demonstrate that our proposed network has

---

[*]Equal Contribution

2nd Workshop on Shared Visual Representations in Human and Machine Intelligence (SVRHM), NeurIPS 2020.

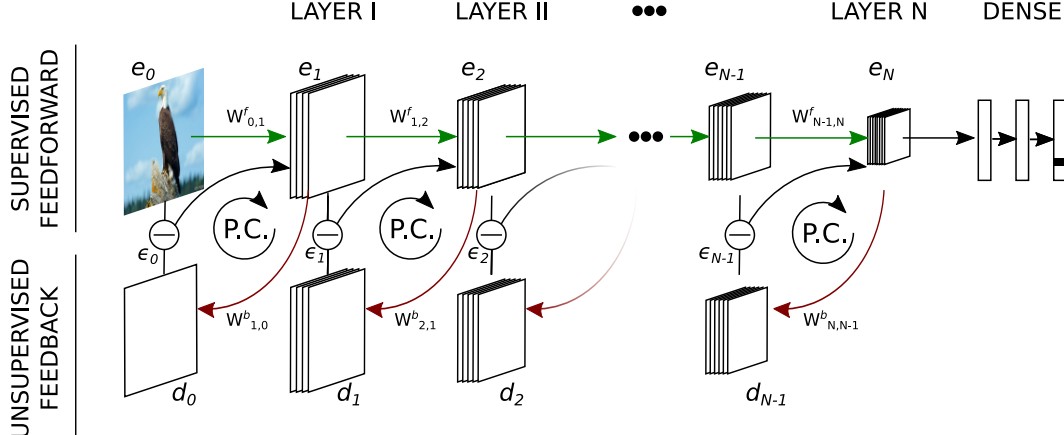

Figure 1: **General overview of our predictive coding strategy** as implemented in a feedforward hierarchical network with generative feedback connections. The architecture (similar to stacked auto-encoders) consists of $N$ encoding layers $e_n$ and $N$ decoding layers $d_n$. $W_{m,n}$ denotes the connection weights from layer $m$ to layer $n$, with $W^f$ and $W^b$ for feedforward and feedback connections, respectively. The reconstruction errors at each layer are denoted by $\epsilon_n$. The proposed predictive coding updates are denoted by 'P.C.' loops. The feedforward connections (green arrows) are trained for image classification (in a supervised fashion), while the feedback weights (red arrows) are optimized for a prediction or reconstruction objective (unsupervised).

interesting properties, especially when viewed from the perspective of robustness. In a separate companion paper [15], we also demonstrate that the proposed dynamics help the networks perceive illusory contours akin to humans. Our contributions can be summarized as follows:

- We propose a novel strategy for effectively incorporating recurrent feedback connections based on the neuroscientific principle of predictive coding.

- We implement this strategy in a feedforward architecture, and show that this improves its robustness against different types of natural and adversarial noise.

- We suggest and verify that the network iteratively shifts noisy representations towards the corresponding clean representations—a form of 'projection towards the learned manifold' as implemented in other strong adversarial defense methods.

## 2 Prior work

Other studies have also tried implementing predictive coding mechanisms in machine learning models [11, 12, 13, 14]. Out of these, our implementation is most similar to the Predictive Coding Networks (PCNs) of [13]. These hierarchical networks were designed with a similar goal in mind: improving object recognition with predictive coding updates. Their proposed dynamics are roughly comparable to ours without the feedforward term, i.e. $\beta_n = 0$. However the most significant difference between their work and ours is that their network (including the feedback connection weights) is solely optimized with a classification objective. As a result, their network does not learn to uniformly reduce reconstruction errors over timesteps (see Figure 4a in the Appendix), as the predictive coding theory would mandate. We also found that their network performs relatively poorly until the final timesteps (see Figure 4b in the Appendix), which does not seem biologically plausible: biological systems typically cannot afford to wait until the last iteration before detecting a prey or predator. We discuss these PCNs [13] further in the Appendix, together with our own detailed exploration of their network's behavior.

Other approaches to predictive coding for object recognition include Boutin et al. [14], who used a PCN with an additional sparsity constraint. The authors showed that their framework can give rise to receptive fields which resemble those of neurons in areas V1 and V2 of the primate brain. They also demonstrated the robustness of the system to noisy inputs, but only in the context of

reconstruction—they did not show that their network can perform (robust) classification, and they did not extend their approach to deep neural networks.

Spratling [16] also described PCNs designed for object recognition, and demonstrated that their network could effectively recognise digits and faces, and locate cars within images. Their update equations differed from ours in a number of ways: they used divisive/multiplicative error correction (rather than additive), and a form of biased competition to make the neurons 'compete' in their explanatory power. The weights of the network were not trained by error backpropagation, making it difficult to scale it to address modern machine learning problems. Indeed, the tasks on which they tested their network are simpler than ours, and the datasets much smaller.

Huang et al. [17] also tried to extend the principle of predictive coding by incorporating feedback connections such that the network maximizes 'self consistency' between the input image features, latent variables and label distribution. The iterative dynamics they proposed, though different from ours, improved the robustness of neural networks against gradient-based adversarial attacks on datasets such as Fashion-MNIST and CIFAR10.

## 3  Our approach

Predictive coding, as introduced by [18], is a neurocomputational theory positing that the brain maintains an internal model of the world, which it uses to actively predict the observed stimulus. Within a hierarchical architecture, each higher layer attempts to predict the activity of the layer immediately below, and the errors made in this prediction are then utilized to correct the higher-layer activity.

To establish our notation, let us consider a hierarchical feedforward network equipped with generative feedback connections, as represented in Figure 1. The network counts $N$ encoding layers $e_n$ and $N$ corresponding decoding layers $d_{n-1}$ (see Figure 1). The feedforward weights connecting layer $n-1$ to layer $n$ are denoted by $W^f_{n-1,n}$, and the feedback weights from layer $n+1$ to $n$ by $W^b_{n+1,n}$. For a given input image, we initiate the activations of all encoding layers with a feedforward pass. Then, over successive recurrent iterations (referred to as timesteps $t$), both the decoding and encoding layer representations are updated using the following equations :

$$d_n(t+1) = \left[W^b_{n+1,n}e_{n+1}(t)\right]_+$$

$$e_n(t+1) = \beta_n\left[W^f_{n-1,n}e_{n-1}(t+1)\right]_+ + \lambda_n d_n(t+1) + (1-\beta_n-\lambda_n)e_n(t) - \alpha_n\frac{\partial\epsilon_{n-1}(t)}{\partial e_n(t)},$$

$$(1)$$

where $\left[\,\cdot\,\right]_+$ is the ReLU function and $\beta_n$, $\lambda_n$, and $\alpha_n$ ($0 \leq \beta_n + \lambda_n \leq 1$) act as layer-dependent balancing coefficients for the feedforward, feedback, and error-correction terms, respectively. $\epsilon_{n-1}(t)$ denotes the reconstruction error at layer $n-1$ and is defined as the mean squared error (MSE) between the representation $e_{n-1}(t)$ and the predicted reconstruction $d_{n-1}(t)$ at that particular timestep. Layer $e_0$ is defined as the input image and remains constant over timesteps. Finally, all the weights $W^f_{n-1,n}$ and $W^b_{n+1,n}$ are fixed during these iterations.

Each of the four terms in Equation 1 contributes different signals: (i) the feedforward term (controlled by parameter $\beta$) provides information about the (constant) input and changing representations in the lower layers, (ii) the feedback term (parameter $\lambda$) conveys information about higher-level representations, as proposed in [19], (iii) the memory term acts as a time constant to retain the current representation over successive timesteps, and (iv) the error correction term (controlled by parameter $\alpha$) fulfills the objective of predictive coding, i.e., it corrects representations in each layer such that their generative feedback can better match the preceding layer. For this error correction term, we directly use the error gradient $\partial\epsilon_{n-1}/\partial e_n$ to take full advantage of modern machine learning capabilities. While the direct computation of this error gradient is biologically implausible, it has been noted before that it is mathematically equivalent to propagating error residuals up through the feedback connection weights $(W^b)^T$, as often done in other predictive coding implementations [18, 13].

While it is certainly possible to train such an architecture in an end-to-end fashion, by combining a classification objective for the feedforward weights $W^f$ with an unsupervised predictive coding

objective (see Section 4) for the feedback weights $W^b$, we believe that the benefits of our proposed scheme are best demonstrated by focusing on the added value of the feedback pathway onto a pre-existing state-of-the-art feedforward network. Consequently, we implement the proposed strategy with an existing feedforward deep convolutional neural network (DCNN) architecture as the backbone: VGG16 trained on ImageNet. We show that predictive coding confers higher robustness to this network.

## 4    Model architecture and training

We start from a VGG16 pretrained on ImageNet and implement its predictive version, PVGG. The network 'body' (without the classification head) is split into a cascade of five sub-modules. Each convolution block is considered as a sub-module which plays the role of an $e_n$ in equation (1). We then add deconvolution as feedback layers $d_{n-1}$ connecting each $e_n$ to $e_{n-1}$, with kernel sizes accounting for the increased receptive fields of the neurons in $e_n$ (see Appendix A.1). We then train the parameters of the feedback deconvolution layers with an unsupervised reconstruction objective (with all other parameters frozen). We minimize the reconstruction errors just after the first forward pass (i.e. no error correction or predictive coding recurrent dynamics are involved at this stage):

$$\mathcal{L} = \sum_{n=0}^{N-1} \| e_n - d_n \|_2, \tag{2}$$

where $N$ is the total number of encoders, $e_n$ is the output of the $n^{th}$ encoder after the first forward pass, and $d_n$ is the estimated reconstruction of $e_n$ via feedback/deconvolution (from $e_{n+1}$).

After training the feedback deconvolution layers, we freeze all of the parameters, and choose the values of hyperparameters $\beta_n$, $\alpha_n$ and $\lambda_n$ for the encoders/decoders update equations (1). These values are chosen manually to decrease the prediction errors over successive iterations. See Appendix A for the chosen hyperparameter values. The resulting model, along with pretrained weights, will be made available at `https://github.com/rufinv/PVGG16-SVRHM2020`.

## 5    Results and Discussion

When considered at time step 0 (i.e., after a single feedforward and feedback pass through the network), the model and its accuracy are—by construction—exactly identical to the standard pretrained VGG16. Over successive time steps, however, the influence of feedback and predictive coding iterations becomes visible. On the standard ImageNet test dataset (clean images), accuracy remains roughly constant over time (Figure 2b brown plot-line); that is, feedback connections do not improve (nor impair) performance. Our hypothesis, on the other hand, is that feedback can confer robustness to image perturbations.

**Gaussian noise robustness:** To evaluate the network robustness, we first inject additive Gaussian noise to the ImageNet test dataset, and quantify the model performance. In terms of accuracy, over successive iterations the PVGG model progressively discards some of the noise and improves its performance relative to the feedforward VGG baseline (see Figure 2b). To understand this robustness to noisy images, we evaluate the quality of image reconstructions generated by the network with MSE (see Figure 2). The reconstructions become progressively cleaner over timesteps. It should be noted that the feedback connections were trained only to reconstruct clean images; therefore, this denoising property is an emerging feature of the model dynamics. Next, we test whether this denoising behavior is also observed in higher layers of the model. To quantify similarity at the level of neural representations rather than pixels, we use the correlation distance. We pass clean and noisy versions of the same image through the network, and measure the correlation distance between the clean and noisy representations at each layer and for each timestep. These correlation distances were normalized with the distance measured at t=0 (i.e., relative to a standard feedforward VGG). The correlation distances decrease consistently over timesteps across all layers of the network (Figure 2d). This implies that feedback predictive coding iterations help the network steer the noisy representations closer to the representations elicited by the corresponding (unseen) clean image.

This is an important property for robustness. When compared to clean images, noisy images can result in different representations at higher layers [20]. Various defenses have aimed to protect

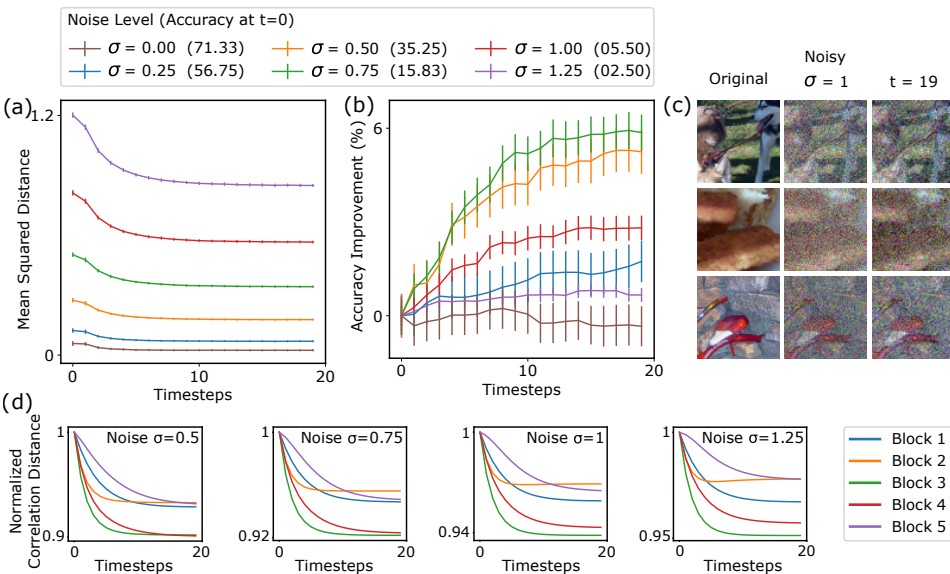

Figure 2: **Predictive coding iterations steer noisy reconstructions and representations towards their learned data manifold:** (a) MSE between clean input images and PVGG's reconstructions when provided with corrupted versions of the same images (additive Gaussian noise with different standard deviations). Over timesteps, reconstructions move closer to the clean version of the input. (b) Improvement in classification accuracy for images corrupted with Gaussian noise, relative to the feedforward VGG baseline (actual accuracy values are listed in the legend). (c) Examples of noisy image reconstructions generated by PVGG. The network only receives noisy inputs (middle) but removes some of the noise after 19 timesteps (right). (d) Encoder-wise normalized correlation distances between representations obtained by clean vs. noisy images. Each subplot corresponds to different levels of Gaussian noise. The correlation distances decrease across timesteps, indicating that the noisy representations become more and more similar to their (unseen) clean version.

neural networks from adversarial attacks by constraining the images to the 'original data manifold'. Accordingly, studies have used generative models such as GANs [21, 22, 23, 24] or PixelCNNs [25] to constrain the input to the data manifold. Similarly, multiple efforts have been made to clean the representations in higher layers and keep them closer to the learned latent space [26, 27, 28, 20]. Here, we demonstrate that feedback predictive coding iterations can achieve a similar goal by iteratively projecting noisy representations towards the manifolds learned during training, both in pixel space and representation spaces.

**Benchmarking robustness:** Finally, we quantify the PVGG model's accuracy on different types of perturbations. We first use ImageNet-C, a test dataset provided by [29] using 19 types of image corruption across 5 severity levels each. Using the recommended mean Corruption Error (mCE) score, we observe a steady improvement of accuracy over timesteps (see Figure 3a and 5), relative to the standard VGG16 feedforward model. We also test the model's robustness with a suite of targeted adversarial attacks. The attack suite consists of gradient-based attacks: BasicIterativeMethod [30], RandomProjectedGradientDescent [31], CarliniWagner attacks [32], and non-gradient based Boundary attacks [33]. The use of a non-gradient based attack ensures that any robustness we observe is not due to *gradient-masking*, a well-known inferior form of defense [34, 35].

Each attack is used against the model, on images that were initially correctly classified, to find a minimal perturbation such that the resulting "adversarial image" is misclassified as another predefined category. Figure 3b-f shows the median perturbation required by each attack to fool the network (see Appendix A.3 for further details). Over successive iterations, increasing perturbations are required, suggesting that predictive coding can help the model to withstand adversarial perturbations.

So far, the most promising strategy for achieving robustness has been adversarial training, whereby adversarial datapoints are added to the training dataset. While efficient, this strategy was also shown to be strongly limited [36, 37]. Most importantly, it shares very little, if any, resemblance to the way the brain achieves robustness. Instead, here we start from biological principles and show that

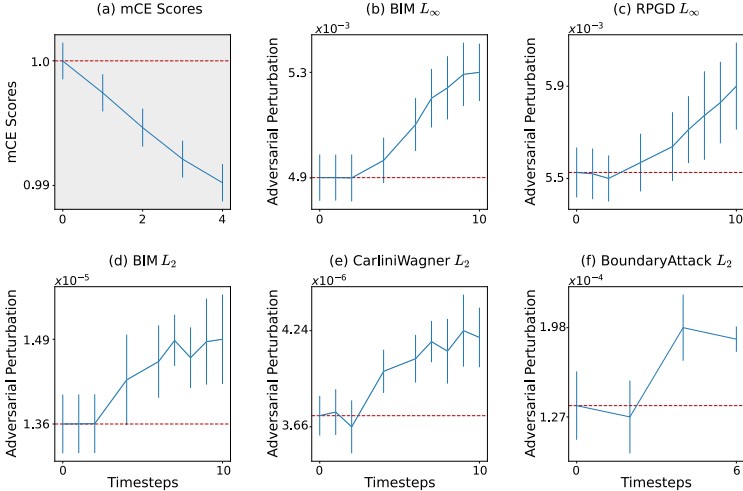

Figure 3: **PVGG robustness to natural and adversarial noise :** (a) mean Corruption Error (mCE) scores calculated on ImageNet-C dataset (19 corruption types, 5 severity levels each) over timesteps (lower is better). mCE scores are normalized to the feedforward VGG mCE score, error bars depict standard deviations from bootstrapping. (b-f) median adversarial perturbations required by different adversarial attacks to successfully fool the network (higher is better). Error bars (obtained from bootstrapping) represent the standard deviation of the medians of the sampled populations. In all panels, the red line denotes the median perturbation value for a feedforward VGG.

they can lead to improved adversarial robustness. It is worth mentioning that PVGG's robustness is achieved here totally via unsupervised training of the feedback connections (while of course, the backbone VGG feed-forward network that we used was pretrained in a supervised manner). We avoid using costly adversarial training, or tuning our hyperparameters specifically for classification under each attack or noise condition. This likely explains why the model, while improving in robustness compared to its feedforward version, remains far from state-of-the-art adversarial defenses. On the other hand, we believe that addition of these methods (adversarial training, hyperparameter tuning) to the training paradigm, in future work, could further improve the network's adversarial robustness.

## 6 Conclusion

In this paper we demonstrated how a state-of-the-art feedforward neural network can be augmented with unsupervised brain-inspired dynamics. Specifically, we showed that predictive coding updates can improve the robustness of the network to a wide variety of random noises, as well as targeted adversarial attacks. We demonstrated that the network achieves this robustness by learning a form of manifold projection. In future work, we intend to generalize these results to other feedforward architectures (e.g. ResNet [38]). We believe that this work contributes to the general case for continuing to draw inspiration from biological visual systems in computer vision, both at the level of model architecture and dynamics.

## Broader Impacts

The research discussed above proposes novel ways of using brain-inspired dynamics in current machine learning models. Specifically, it demonstrates a neuro-inspired method for improving the robustness of machine learning models. It further highlights one of the many important properties necessary for obtaining this robustness. Given that such models are employed by the general public, and are simultaneously shown to be heavily vulnerable, research efforts to increase (even marginally) or to understand their robustness against mal-intentioned adversaries has high societal relevance.

Importantly, the research also aims to bridge techniques between two different fields–neuroscience and machine learning, which can potentially open new avenues for studying the human brain. For example, it could help better understand the unexplained neural activities in patients, to improve their living conditions, and in the best case, in the treatment of their conditions. While this may also be associated with inherent risks (related to privacy or otherwise), there are clear potential benefits to society.

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

# A    Appendix

## A.1    Network Architectures

Here we provide the details on how VGG16 is converted to a predictive coding network–PVGG.

VGG16 consists of five convolution blocks and a classification head. Each convolution block contains two or three convolution+ReLU layers with a max-pooling layer on top. For each $e_n$ in PVGG, we selected the max-pooling layer in block $n-1$ and all the convolution layers in block $n$ of VGG16 (for $n \in \{1, 2, 3, 4, 5\}$) as the sub-module that provides the feedforward drive. Afterwards, for each $e_n$ a deconvolution layer $d_{n-1}$ is added which takes the $e_n$ as the input and reconstructs (or predicts) the output of $e_{n-1}$. We call a bundle of $(e_n, d_{n-1})$ as a Predictive Coder module ($PC_n$). Please note that we consider $e_0$ as the input image. Table 1 summarizes PVGG's architecture.

The hyperparameter values that we used are provided in Table 2.

Table 1: Architectures of $e_n$s and $d_n$s for PVGG. Conv (channel, size, stride), MaxPool (size, stride), Deconv (channel, size, stride)

| | PVGG | |
| --- | --- | --- |
| | $e_n$ | $d_{n-1}$ |
| $PC_1$ | $\big[\text{Conv }(64, 3, 1)\big]_+$ 
 $\big[\text{Conv }(64, 3, 1)\big]_+$ | Deconv (3, 5, 1) |
| $PC_2$ | MaxPool (2, 2) 
 $\big[\text{Conv }(128, 3, 1)\big]_+$ 
 $\big[\text{Conv }(128, 3, 1)\big]_+$ | $\big[\text{Deconv }(64, 10, 2)\big]_+$ |
| $PC_3$ | MaxPool (2, 2) 
 $\big[\text{Conv }(256, 3, 1)\big]_+$ 
 $\big[\text{Conv }(256, 3, 1)\big]_+$ 
 $\big[\text{Conv }(256, 3, 1)\big]_+$ | $\big[\text{Deconv }(128, 14, 2)\big]_+$ |
| $PC_4$ | MaxPool (2, 2) 
 $\big[\text{Conv }(512, 3, 1)\big]_+$ 
 $\big[\text{Conv }(512, 3, 1)\big]_+$ 
 $\big[\text{Conv }(512, 3, 1)\big]_+$ | $\big[\text{Deconv }(256, 14, 2)\big]_+$ |
| $PC_5$ | MaxPool (2, 2) 
 $\big[\text{Conv }(512, 3, 1)\big]_+$ 
 $\big[\text{Conv }(512, 3, 1)\big]_+$ 
 $\big[\text{Conv }(512, 3, 1)\big]_+$ | $\big[\text{Deconv }(512, 14, 2)\big]_+$ |

Table 2: Values of the Hyperparameters

| $n$ | $\beta_n$ | $\lambda_n$ | $\alpha_n$ |
| --- | --- | --- | --- |
| 1 | 0.2 | 0.05 | 120 |
| 2 | 0.4 | 0.10 | 0.250 |
| 3 | 0.4 | 0.10 | 0.020 |
| 4 | 0.5 | 0.10 | 0.004 |
| 5 | 0.6 | 0.10 | 0.010 |

## A.2    Prior work: PCNs

To better understand the model proposed by Wen et al. [13] and its differences to ours, we conducted several experiments using the code that they provided, and report here our most compelling observations. A first striking shortcoming was that the accuracy of their feedforward baseline was far from optimal. Using their code, with relatively minor tweaks to the learning rate schedule, we were able to bring it up from  60% to 70% – just a few percentage points below their recurrent network. We expect that this could be further improved with a more extensive and systematic hyperparameter search. In other words, their training hyperparameters appeared to have been optimised for their predictive coding network, but not – or not as much – for their feedforward baseline. We further found that a minor change to the architecture - using group normalisation layers after each ReLU – leads to a feedforward network which performs on par with the recurrent network, with a mean over 6 runs of 72% and best of 73%. Adding the same layers to the recurrent network did not lead to a corresponding improvement in accuracy.

We also found that the network had poor accuracy (underperforming the optimized feedforward baseline) until the final timestep, as can been seen in Figure 4b. This can be clarified by a closer

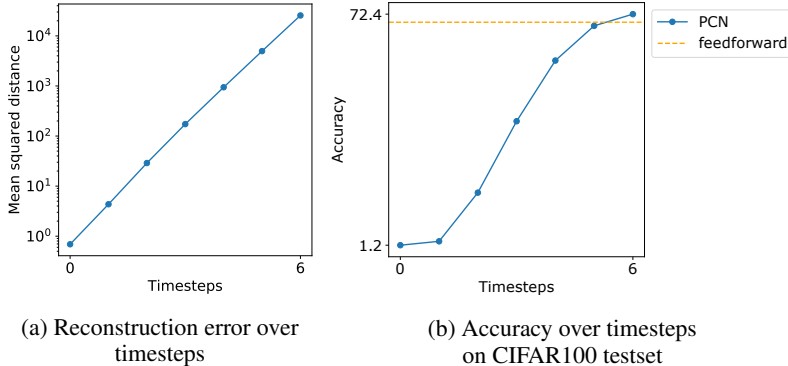

(a) Reconstruction error over timesteps

(b) Accuracy over timesteps on CIFAR100 testset

Figure 4: **PCN**: Panel (a) shows the reconstruction errors of the model over timesteps. It does not decrease over timesteps, as would be expected in a predictive coding system. Panel (b) depicts the accuracy of the model on the CIFAR100 test dataset. The model performs at chance level at early timesteps and then becomes better in the last few timesteps.

reading of Figure 3 of their paper: the reported improvements over cycles from 60% at time step 0 to more than 70% at time step 6 are for seven distinct networks, each evaluated only at the timestep they were trained for. So in fact, in their model the predictive coding updates do not gradually improve on an already reasonably guess. This is clearly not biologically plausible: visual processing would be virtually useless if the correct interpretation of a scene only crystallised after a number of 'time steps'. By the time a person has identified an object that object is likely to have disappeared or, in a worst case scenario, eaten them. We also experimented with feeding the classification error at each timestep into an aggregate loss function, but this lead to a network which, while performing well, essentially did not improve over timesteps.

Figure 4a shows that the network does not uniformly minimise reconstruction errors over time for all layers, and thus is not performing correct predictive coding updates. In fact the total reconstruction error (across all layers) increases exponentially over timesteps. There are a number of possible explanations for this. Firstly, in the case of the network with untied weights, the authors choose to make a strong assumption in the update equations (seen as the equivalence of their Equations 5 and 6): that the feedback weights can be assumed to be the transpose of the feedforward weights, i.e. $W^b = (W^f)^T$. They thus propagate the feedforward error through the feedforward weights. However, it might be that the network learns feedback weights which essentially invert the feedforward transformation as assumed, but this is not guaranteed, and nor is it explicitly motivated through the classification loss function. Indeed, because the network is not motivated to learn a representation at earlier timesteps which produces a good prediction, it does not necessarily need to learn the inverse transformation: it can instead learn some other transformation which, when applied with the update equations, leads the network to *end up* in the right place. That being said, this assumption is valid for the network with tied weights, and this network also does not uniformly reduce reconstruction error over timesteps. Possibly, the presence of ReLU non-linearities means that the forward convolution may still not be perfectly invertible by a transposed convolution. Finally, in line with this unexpected increase of reconstruction errors over time, we have also failed to extract good image reconstructions from the network as seen in Figure 5 of their paper, although in private communication the authors indicated that this was possible with some other form of normalisation.

In short, while the ideas put forward in [13] share similarities with our own, their exact implementation did not support the claims of the authors, and the question of whether predictive coding can benefit deep neural networks remained an open one. We hope that our approach detailed in the present study can help resolve this question.

## A.3 Details on adversarial attacks

Adversarial attacks are hard-mined perturbations that aim to change the classification label of a network with little perturbation. Here, we use a suite of targeted attacks, which aim to change the classification label to a strictly predefined category. The suite included iterative attacks that relied on gradient information (BasicIterativeMethod [30], RandomProjectedGradientDescent [31],

CarliniWagner [32]) and one attack that did not rely on gradient information (BoundaryAttack) [33]. Attacks like boundary attacks are used to ensure that the defense does not utilize a technique known as *gradient-masking*, wherein a defense knowingly or unknowingly relies on hiding the gradients from the attacker. Stronger attacks that can estimate the gradients in the target network can easily break such defenses [34].

To allow for a meaningful comparison across timesteps, we add an additional criteria that any image used has to be correctly classified by the model on all the timesteps chosen for attack. For boundary attacks, since we need to use a target image (instead of a target-class), we also enforce this criteria for target images and use the same target image across all timesteps. The numbers for attacks BIM $L_\infty$, RPGD $L_\infty$, BIM $L_2$, CarliniWagner $L_2$ are 500, 127, 165, and 112 respectively for all timesteps. These numbers were restricted mainly by computational requirements. For boundary attacks, the numbers are 209, 72, 98 and 95 for timesteps 0, 2, 4, and 6 respectively. All the attacks were conducted using Foolbox 2.3.0 API. The adversarial perturbations are then used as a base population to bootstrap and estimate the median perturbation. The standard deviation of the medians is denoted as error bars in Figure 3.

## A.4 CE Scores

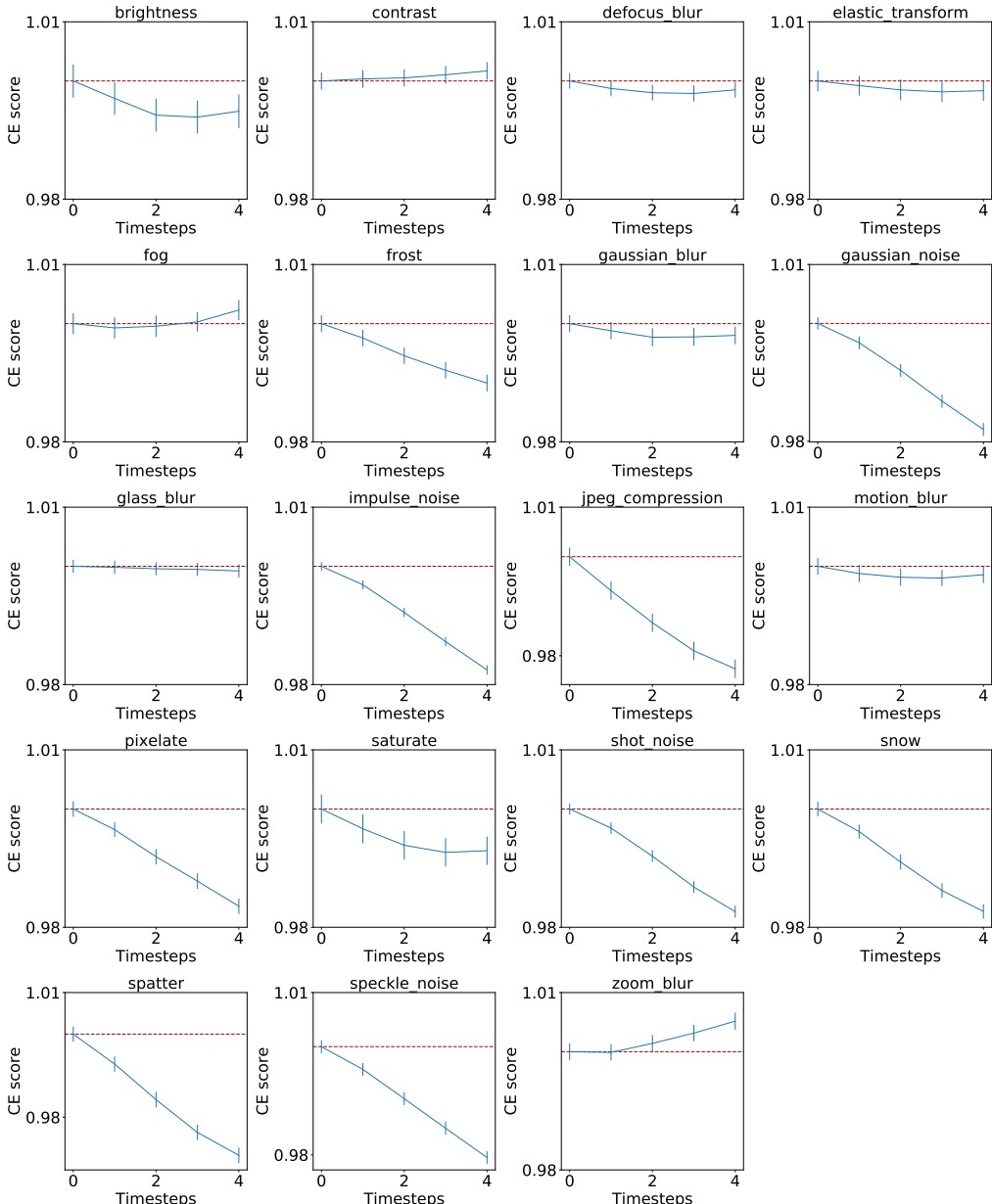

Figure 5: **Corruption Error (CE) scores for all distortions:** The panel shows the CE scores calculated on the distorted images provided in the ImageNet-C dataset. The values are normalized with the CE score obtained for the feedforward VGG. The error bars denote the standard deviation of the means obtained from bootstrapping (resampling multiple binary populations across all severities.)

