# OpenReview forum: "Brain-inspired predictive coding dynamics improve the robustness of deep neural networks"
_NeurIPS.cc/2020/Workshop/SVRHM — SVRHM@NeurIPS Poster_

### Official Review · AnonReviewer2 · 2020-10-28
**Important problem and interesting results, however unclear impact on traditional classification performance**

**Rating:** 7
**Confidence:** 4

**Review:**

The problem that the paper addresses is an important and an interesting one. The role of feedback in networks is still disputed and this paper provides some evidence towards a solution. The approach is fairly straightforward yet provides some interesting results. Though there are still many open questions remaining, such as how would this model handle temporal sequences, this nonetheless provides an interesting start.

The model is also shown to be robust to a number of types of adversarial attacks. However, one of the major concerns and limitations of the paper is that the model and its dynamics have not been tested under the traditional supervised classification setting. It is not clear these feedback connections help with traditional classification error minimization.  Though it is impressive that robustness increases with timesteps through the network, it is unclear whether classification noise such as due to intra class nuisance variation can be handled better using such a predictive feedback mechanism?

---

### Official Review · AnonReviewer3 · 2020-10-29
**A modest step forward towards biologically plausible adversarial robustness**

**Rating:** 8
**Confidence:** 3

**Review:**

This is a timely attempt to achieve biologically-plausible robust classification by introducing predictive-coding recurrent dynamics to a pretrained convolutional deep neural network. Adversarial examples pose a difficult challenge to convolutional deep neural networks as models of human recognition. The leading CS-based solution (adversarial training) has no biological plausibility, and it operates by artificially extending the training data instead of introducing a better inductive bias to the model. Therefore, the aim of the current work is very significant.

However, I am not sure that this particular implementation (augmenting a fixed feedforward VGG with top-down predictive coding connections) is sufficient for achieving this tall order aim. Unlike the Rao & Ballard 1999 model or its supervised adaptation by Spratling (2017, Cognitive Computation), the training of the weights of the proposed model is not governed by a generative objective. The feedforward connections are not finetuned to support more predictive high-level representation, and the top-down connections only learn to predict this fixed, pretrained representation. Therefore, this network should be conceived and presented as a predictive coding-inspired model rather than a proper implementation of the Bayesian predictive coding approach. This divergence from Bayesian predictive coding may (or may not) explain the qualitatively modest improvements achieved in model robustness. Having said that, I agree with the authors that their work is a step forward from the Wen 2018 PCN model, and I think that it is a step in the right direction. I therefore recommend this paper for presentation in SVRHM.

#### Additional suggested points for improvement down-the-road:
 1) Introduction: in my opinion, predictive coding is not supported by `ample’ neuroscience evidence. It is actually quite debated. A few balanced reviews on the empirical evidence for predictive coding that can be cited in this context are Heilbron & Chait, 2018 Neuroscience; Aitchison & Lengyel, 2017 Curr Opin Neurobiol; Walsh, McGovern, Clark & O’Connell, 2020 Ann N Y Acad Sci.
 2) As mentioned above, the model is *not* equivalent to a supervised extension of the Rao & Ballard 1999 model. It would be illuminating if the authors could motivate their proposed model by starting from the classical predictive coding model (i.e., Rao & Ballard) and then explain how (and why) they modified its underlying assumptions to arrive at equations (1) and (2).
 3) Apply a formal hyperparameter search for $\beta_n$ $\lambda_n$ and $\alpha_n$ instead of manual tuning; Explicitly report the criterion used for hyper-parameter choice, including the cross-validation scheme.
 4) Robustness evaluation (all panels of figure 2) lacks context and comparison to alternative models. These should include adversarially trained CNNs as well as competing implementations of predictive-coding based classification.
 5) Test and discuss the effect of minimizing the error of the first timepoint vs. a temporal average of the error.
 6) Cite and compare to recent similar works (e.g., Huang et al., 2020, arXiv:2007.09200).

#### Minor points:
 7) Reporting of accuracy (Figure 2 panel b): a relative scale can be used in addition to an absolute scale, not instead of it.
 8) Reporting of correlation distance (Figure 2, panel d): the y-axis label should be ‘Normalized Correlation Distance’ and not just ‘Correlation Distance’. The normalization should be defined and discussed in the text.

---

### Public Comment · ~Bhavin_Choksi1 · 2020-12-05
**General response to reviewers**

We would like to thank all the reviewers for their insightful and valuable comments. All the authors agree with the points raised by the reviewers, in particular regarding the use of unsupervised (AnonReviewer3) or supervised (AnonReviewer2) training objectives (for our feedback and feedforward connection weights, respectively). We agree that future experiments should aim at methodically exploring these choices, to further our understanding of recurrent connections within the proposed predictive coding framework. This work is ongoing, and we expect to report the corresponding findings as an independent, fully detailed paper. In the meanwhile, we have revised the current manuscript to address many of the reviewers’ outstanding comments.

---

### Decision · Program_Chairs · 2020-11-02

Accept (Poster)